

# Pelvic floor pressure distribution profile in urinary incontinence: a classification study with feature selection

Adriano Carafini[1,*], Isabel C.N. Sacco[2] and Marcus Fraga Vieira[1,*]

[1] Bioengineering and Biomechanics Laboratory, Universidade Federal de Goiás, Goiânia, Goiás, Brazil
[2] Physical Therapy, Speech and Occupational Therapy Department, School of Medicine, Universidade de São Paulo, São Paulo, São Paulo, Brazil
* These authors contributed equally to this work.

Corresponding author
Marcus Fraga Vieira, marcus@ufg.br, marcus.fraga.vieira@gmail.com

## ABSTRACT

**Background**. Pelvic floor pressure distribution profiles, obtained by a novel instrumented non-deformable probe, were used as the input to a feature extraction, selection, and classification approach to test their potential for an automatic diagnostic system for objective female urinary incontinence assessment. We tested the performance of different feature selection approaches and different classifiers, as well as sought to establish the group of features that provides the greatest discrimination capability between continent and incontinent women.

**Methods**. The available data for evaluation consisted of intravaginal spatiotemporal pressure profiles acquired from 24 continent and 24 incontinent women while performing four pelvic floor maneuvers: the maximum contraction maneuver, Valsalva maneuver, endurance maneuver, and wave maneuver. Feature extraction was guided by previous studies on the characterization of pressure profiles in the vaginal canal, where the extracted features were tested concerning their repeatability. Feature selection was achieved through a combination of a ranking method and a complete non-exhaustive subset search algorithm: branch and bound and recursive feature elimination. Three classifiers were tested: k-nearest neighbors (k-NN), support vector machine, and logistic regression.

**Results**. Of the classifiers employed, there was not one that outperformed the others; however, k-NN presented statistical inferiority in one of the maneuvers. The best result was obtained through the application of recursive feature elimination on the features extracted from all the maneuvers, resulting in 77.1% test accuracy, 74.1% precision, and 83.3 recall, using SVM. Moreover, the best feature subset, obtained by observing the selection frequency of every single feature during the application of branch and bound, was directly employed on the classification, thus reaching 95.8% accuracy. Although not at the level required by an automatic system, the results show the potential use of pelvic floor pressure distribution profiles data and provide insights into the pelvic floor functioning aspects that contribute to urinary incontinence.

## INTRODUCTION

Machine learning (ML) methods have the ability to learn about a system's behavior directly from its observed data and do not require previous knowledge on the mathematical relations ruling it. In the biomedical engineering and clinical applications field, these methods have been increasingly employed in the construction of computer-aided diagnosis (CAD) systems. Such systems aim to reduce diagnostic dependence on professionals' experience and consequently the variability and subjectivity of the results (*Mumtaz et al., 2017*; *Kao & Wei, 2011*).

The first stage is determining which features are to be used, followed by choosing and applying the ML methods for CAD system construction. Since CAD systems' target responses are most commonly categorical, a group of ML methods named classifiers perform the modeling task. Frequently chosen classifiers for clinical and biomedical engineering contexts are artificial neural networks, support vector machines (SVMs), decision trees, logistic regression (LR), and random forests (*Shaikhina et al., 2017*).

Using previous knowledge of the biological phenomenon and feature extraction techniques, features are extracted from the raw data and fed to the classifier; these features have as much of an effect on the CAD system's performance as the classifier itself (*Krishnan & Athavale, 2018*). To improve this performance, dimensionality reduction methods are also usually applied between the extraction and classification stages. The reduction may be accomplished through transformation, such as retaining the most relevant components from a principle components analysis (PCA) as well as through feature subset selection (*Krishnan & Athavale, 2018*; *Webb, 2002*). In addition to performance improvement, feature set reduction may remove irrelevant features, hence decreasing the final classifier's complexity (*Webb, 2002*).

In this context, automatic diagnosis of female urinary incontinence (UI) is an example of a CAD system in the clinical setting that is already approached through ML methods. Categorical clinical variables have been used to evaluate different classifiers (genetic algorithm, k-means, LR, and decision trees) for the task of automatically diagnosing three different types of UI (*Laurikkala et al., 1999*).

The International Continence Society defines UI as the complaint of any involuntary leakage of urine, which is further classified into three main subtypes: stress UI, urgency UI, and mixed UI. Regardless of the type, UI negatively affects women's quality of life. Stress UI, for instance, may become an obstacle for regular physical activities and may negatively affect sexual function, thus jeopardizing women's general health and well-being (*Caetano, Tavares da & De Lopes, 2007*; *Nygaard et al., 2015*; *Lim et al., 2016*).

It is well established that the continence mechanism depends on the integrity of passive (conjunctive tissues) and active structures (muscles) within the pelvic floor. Due to time (or trauma), part of these structures can be impaired, leading to the need of rehabilitation approaches to restore the continence function and knowing where and how this forces/pressures generation and maintenance are impaired guide the process of pelvic floor interventions in physical therapy. The pelvic floor muscles (PFM) play an important (twofold) role in maintaining urinary continence: first, by supporting pelvic organs and
restricting bladder neck displacements (*Bø, 2004*); second, by compressing the urethra distally, causing the urethral pressure to increase prior to and during effort tasks, thereby preventing urine leakage (*Miller et al., 2001*; *DeLancey, 1988*). To date there is Level I evidence, Grade A that PFM training is effective in the treatment of women with stress and mixed UI, being recommended as first-line conservative management for UI in women of all ages (*Dumoulin, Cacciari & Hay-Smith, 2018*; *Dumoulin et al., 2017*; *Nambiar et al., 2018*).

Up till now, there is no definition of an optimal force/pressure generation necessary to clamp the urethra and prevent urine leakage or a clear distinction between continent and incontinent women regarding their PFM function, leading to the assumption that being able to discriminate the source of the forces acting on the vaginal canal is as important as its magnitude for the continence mechanism. Studies have already showed lower rest intravaginal pressure or forces in women with UI compared to a continent control group (*Shishido et al., 2008*; *Morin et al., 2004*). However, this is not a consensus in the literature (*Devreese et al., 2004*; *Verelst & Leivseth, 2007*; *Chamochumbi et al., 2012*). Therefore, techniques to discriminate women with and without PF dysfunctions seems to be important to guide further studies and interventions. This was one of our aims within this study and not to specifically and directly applied in the clinical scenario yet.

Overall, we were able to show the intravaginal pressure profile of continent and incontinent women, which was region-dependent, with a pressure pattern that varied between groups and across the length of the vaginal cavity. The lower pressure maintenance precisely at the main PFM action point observed in the incontinent group reinforces the lack of endurance capacity of the PFM this population.

Furthermore, expenditure estimates regarding UI treatment in several countries have demonstrated the substantial economic burden that will continue to increase with an aging population (*Milsom et al., 2014*). Therefore, any contribution that facilitates the diagnosis of UI in women is of interest.

The present work aims to verify the potential for a CAD system to discriminate between continent and incontinent women (female UI), in which the raw input consists of multidimensional intravaginal pressure profiles acquired through a novel instrumented non-deformable probe (*Cacciari et al., 2017a*). With feature extraction being guided by previous knowledge on vaginal pressure profiles related to pelvic floor functionality as well as branch and bound and recursive feature elimination algorithms performing feature selection, we also seek to determine the feature set with greatest potential for discrimination between continent and incontinent women.

## MATERIALS & METHODS

### Data acquisition

A fully instrumented non-deformable probe with capacitive transducers was used to acquire the vaginal spatiotemporal pressure distribution. The probe consisted of an Ertacetal cylinder (tensile modulus of elasticity = 2,800 MPa) covered by a $10 \times 10$ matrix of individually calibrated capacitive sensors (MLA-P1, Pliance System; novel; Munich,

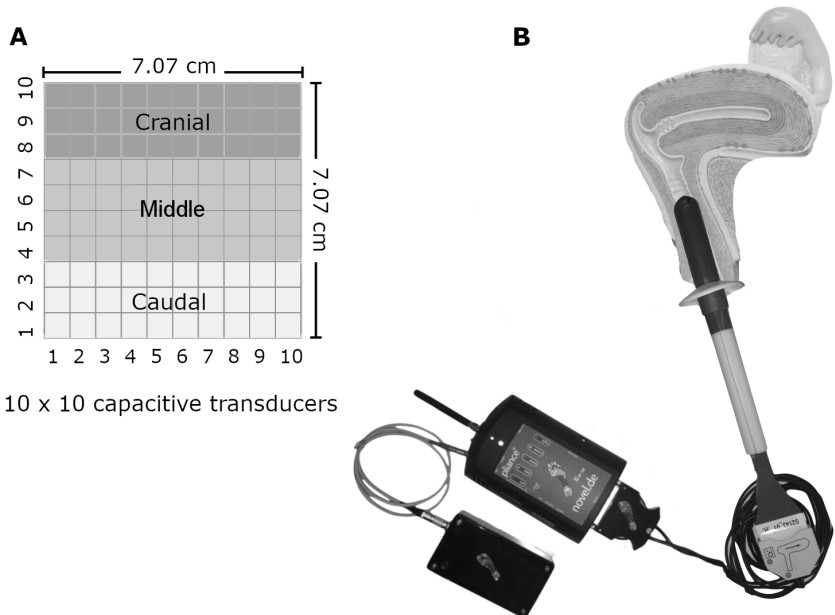

**Figure 1** **The probe and the matrix of pressure sensors.** (A) The dimension and disposition of the pressure sensors on the probe. (B) The fully instrumented probe and its data conditioner positioned within the vaginal canal. Image and photo credit: Isabel Sacco.

Germany). The cylinder was 23.2 mm in diameter and eight cm in length, and its sensing area was $70.7 \times 70.7$ mm ($10 \times 10$ sensing elements of $7.07 \times 7.07$ mm, with 1.79 mm gaps between them) (Fig. 1). The capacitive sensors had a measurement range of 0.50–100.00 kPa and a resolution of 0.42 kPa. Reliability and testing capacities of the instrument were described by (*Cacciari et al., 2017a*; *Cacciari et al., 2017b*).

We divided the sensor matrix into various sections. The first was five planes with 20 sensors each. Each plane (36° apart from each other) represented a sum vector of pressures from two 10-sensor lines diametrically opposed along the cylinder. The second was 10 rings, with each one created by the 10-sensor perimeters surrounding the cylinder. The third division had three lines, each with a 10-sensor perimeter surrounding the cylinder: cranial (corresponding to the first three lines of sensors from the vaginal opening), medial (four mid-lines of sensors), and caudal (three last lines of sensors). The final division was left, posterior, right, and anterior sections. Figure 2 illustrates the instrument with the sensor matrix disposition and a scheme of the sensor subsets.

Data acquisition was performed on 24 adult continent women $35.3 \pm 10.0$ years $23.4 \pm 4.2$ kg/m$^2$) and 24 diagnosed with stress UI $48.2 \pm 8.1$ years $27.5 \pm 3.6$ kg/m$^2$; incontinence impact 66.7/100; severity measures 41.7/100 on the King's Health Questionnaire). To be eligible, women had to be continent or urinary incontinent, not virgins, with no history of pregnancy within the past year, in premenopausal status with monthly menstrual cycles, with a body mass index (BMI) lower than 30 kg/m$^2$, and with no history of pelvic floor muscle (PFM) training or of any medical conditions that could interfere with PFM function. During clinical evaluation, participants were excluded if they
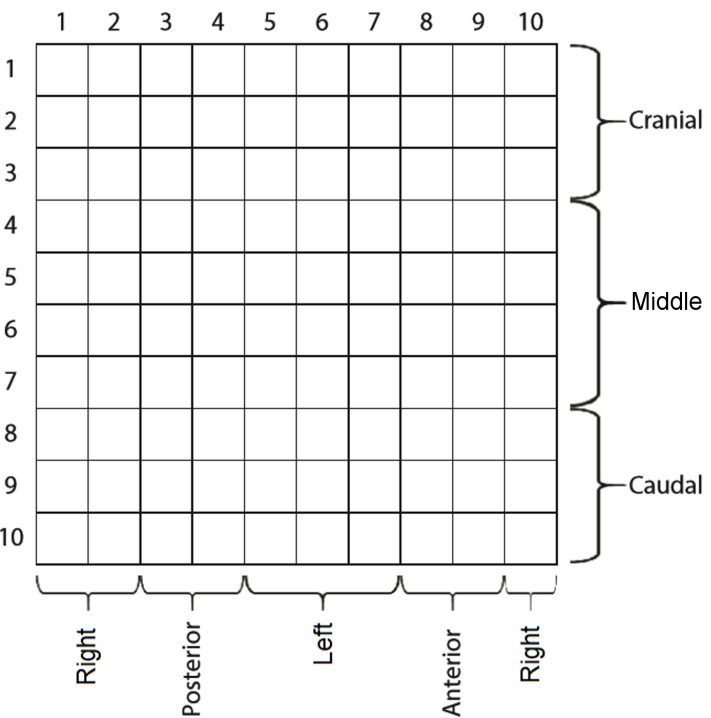

**Figure 2** **The matrix of capacitive sensors on the intravaginal pressure probe.** The probe was carefully introduced so that sensor columns 8 and 9 were anteriorly positioned.

presented above Stage II on the Pelvic Organ Prolapse Quantification (POPQ) scale (*Haylen et al., 2016*) and if they were not able to voluntarily contract their PFM. The King's Health Questionnaire was used to classify the participants into the continent or incontinent group (*Tamanini et al., 2003*). Women reporting no symptoms of any type of UI were included in the control group. This study was approved by the Ethics Committee of the School of Medicine of the University of São Paulo (protocol n.023/14), and all participants provided written informed consent prior to participation.

The probe was always inserted with the same orientation and at a depth of seven cm from the hymenal caruncle according to references marks. After a 1-minute accommodation period, the participants were asked to accomplish four maneuvers (in the same order) with a 1-minute rest between them: (1) the maximum contraction maneuver, in which the participants had to lift and squeeze their pelvic floor as hard as possible for 3 s (*Cacciari et al., 2017a*); (2) the Valsalva maneuver, which consists of executing maximum intra-abdominal pressure effort leading to a downward movement of the pelvic floor for 5 s (*Cacciari et al., 2017a*); (3) the endurance maneuver, in which participants had to sustain pelvic floor contraction for 10 s while breathing normally (*Cacciari et al., 2017b*); and (4) the wave maneuver, in which women were instructed to contract their PFMs in a caudal-cranial direction for 2 s and then relax them in a cranial-caudal direction for 2 s (*Cacciari et al., 2017b*). All participants received standardized verbal support to encourage them to perform maximal PFM contractions throughout the maximal and endurance maneuvers.

The start of the data acquisition was manually synchronized a few seconds before the verbal command. The sampling frequency during pressure data acquisition was set at 50 Hz (*Cacciari et al., 2017a*).

## Feature extraction

Feature extraction was based in previous studies (*Cacciari et al., 2017a*; *Cacciari et al., 2017b*) where the extracted features were capable of distinguishing vaginal sub-regions, planes, rings and maneuvers, important aspects in pelvic floor assessment, and presented excellent inter- and intra-rater reliability and intra-trial repeatability.

Prior to any feature extraction, each sensor time series was filtered by a zero-lag, 8th order, low-pass, Butterworth filter with a cut-off frequency of 8 Hz. Then, the variable extraction process was performed over sensor sets, which were defined either by a maximum or by a sum operator, leading to two different time series: a peak pressure (Eq. 2.1) and a sum pressure time series (Eq. 2.2):

$$Y^S_{peak}[t] = \max_{s'} S[t] \tag{2.1}$$

$$Y^S_{sum}[t] = \sum_{s' \in S[t]} S[t] \tag{2.2}$$

where $S[t]$ is the pressure reading at time instant $t$ of a sensor set $S$, and $s'$ is an element of this set.

For the maximum contraction and Valsalva maneuvers, the extracted variables were the maximum pressure (Eq. 2.3), maximum sum (Eq. 2.4), instant of maximum pressure (Eq. 2.5), instant of maximum sum (Eq. 2.6), and instant of activation (Eq. 2.7):

$$max_p = \max Y^S_{peak}[t] \tag{2.3}$$

$$max_s = \max Y^S_{sum}[t] \tag{2.4}$$

$$t_{max_p} = \operatorname*{argmax}_t Y^S_{peak}[t] \tag{2.5}$$

$$t_{max_s} = \operatorname*{argmax}_t Y^S_{sum}[t] \tag{2.6}$$

$$t_{activ} = t \,|\, Y^S_{peak}[t] > \delta \tag{2.7}$$

where $\delta$ is a threshold corresponding to twice the standard deviation of the peak pressure time series base value (*Cacciari et al., 2017b*).

All variables were computed over six distinct supersets of sensor groupings: $S_{long}$, $S_{lat}$, plane, ring, $S_{long} \cap S_{lat}$, and whole matrix. The first superset, $S_{long}$, includes the posterior, anterior, right, and left groupings (*Guaderrama et al., 2005*), exactly as depicted in Fig. 2. $S_{lat}$ includes the cranial, medial, and caudal groupings (*Guaderrama et al., 2005*); hence, $S_{long} \cap S_{lat}$ is the superset containing the results of the intersection between $S_{long}$ and $S_{lat}$. The remaining two supersets include planes (Eq. 2.8) and rings (Eq. 2.9) of the sensors (*Cacciari et al., 2017a*):

$$Plane_i = C_i \cup C_{i+5} \quad (2.8)$$

$$Ring_j = R_j \quad (2.9)$$

where $C_i$ is the group of sensors corresponding to the $i$-th column and $i \in [1, 5]$ interval. $R_j$ is the group of sensors corresponding to the $j$-th row and $j \in [1, 10]$ interval.

Aside from the features extracted on the maximum contraction and Valsalva maneuvers, three other variables were extracted for the endurance maneuver. The integral of pressure (Eq. 2.10) and integral of sum (Eq. 2.11) were computed over the Plane and Ring supersets, while the plateau duration (Eq. 2.12) was computed only for the whole matrix of sensors (*Cacciari et al., 2017b*):

$$int_p = \sum_t Y^S_{peak}[t] \quad (2.10)$$

$$int_s = \sum_t Y^S_{sum}[t] \quad (2.11)$$

$$\Delta_{plateau} = \max\left(\Delta t \,|\, Y^S_{peak}[t] \geq 0.9 \cdot max_p, \forall t \in \Delta t\right) \quad (2.12)$$

where $\Delta t$ is any time interval within the maneuver execution.

In the wave maneuver, the feature's maximum pressure, instant of maximum pressure, integral of pressure, integral of sum, and instant of activation were extracted from all six aforementioned supersets. Moreover, the rate of contraction (Eq. 2.13) and rate of relaxation (Eq. 2.14) were computed only for the whole matrix (*Cacciari et al., 2017b*):

$$CR = \frac{\max Y^S_{peak}\left[t'\right] - Y^S_{peak}[0]}{\underset{t'}{\mathrm{argmax}} Y^S_{peak}\left[t'\right]} \quad (2.13)$$

$$RR = \frac{\max Y^S_{peak}[1] - \max Y^S_{peak}\left[t'\right]}{1 - \underset{t}{\mathrm{argmax}} Y^S_{peak}\left[t'\right]} \quad (2.14)$$

where $t'$ is the normalized time of the interval $[0, 1]$.

Finally, sample covariances were computed from the sensor groupings pertaining to the $S_{long} \cap S_{lat}$ superset for the Valsalva, maximum contraction, and endurance maneuvers. The elements of the main and inferior diagonals of the sample covariance matrix were used as features.

## Feature selection

The applied process of feature selection was composed of a ranking stage followed by a complete feature subset search (FSS), both independent of the classification process. Two FSS algorithms were tested: extended branch and bound algorithm (BB) and recursive feature elimination algorithm (RFE).

The extended BB FSS algorithm (*Duc & Andrianasolo, 1998*), with the exception of the functionality of determining variables to be disregarded in the search process, was implemented in the Python programming language using the Mahalanobis distance as the subset evaluation metric. This algorithm, despite performing a complete FSS without exhausting every feature combination, may imply a high computational cost since both the number of evaluated subsets and dimension of the covariance matrix to be inverted during the Mahalanobis distance calculation grow exponentially as a function of the number of features (*Webb, 2002*).

Given this computational burden, prior to the application of the BB algorithm, the N most relevant features (ranked according to the Pearson and RELIEF criteria) were kept for FSS. Incremental values of N were evaluated using the execution time as a stopping criterion, thus resulting in the value $N = 25$. Finally, in a similar procedure but with the mean accuracy as the stopping criteria, the range of dimensions of the subsets selected by the BB method was determined to be (*Kao & Wei, 2011*; *Bø, 2004*).

The Recursive Feature Elimination (RFE) algorithm iteratively fits a model, computes the model dependent ranking criterion and discards the last M features ranked by the criterion. This model dependency implies that the top ranked features are not necessarily the ones that are individually most important, thus working as a feature subset ranking method. The base model used to compute the criterion was the Random Forest, with the feature importance being its criterion. The model commonly used for this FSS method, the SVM, was not used since it is already part of the evaluated models, and it could create bias in its favor. Finally, the RFE parameters, as well as random forest parameters, both provided by the python library *scikit-learn*, were selected using an unreplicated regression analysis, with the target being the test accuracy, having *rfe_step* (M features to remove at a time), *n_estimators*, *min_samples_split* and *max_features* as 2 levels factors. The activeness of which factor was evaluated used Lenth method (*Lenth, 2008*), from which none was deemed active. This way, the sign of the coefficients was used to determine the values of each factor: *rfe_step* at 5, *n_estimators* at 10, *min_samples_split* at 2 and *max_features* at 0.5.

## Classifiers configuration, selection, and evaluation

The Python package scikit-learn was used to implement the classifiers k-nearest neighbors (k-NN), LR, and SVM, which were evaluated on the classification of the intravaginal pressure data. The higher the numbers of hyperparameters to be explored, the higher the chances of overfitting due to configuration selection process, compromising the classification performance over independent samples (*Cawley & Talbot, 2010*). Thus, the number of hyperparameters with scanning range was fixed to 1 for each model.

For the k-NN classifier, the hyperparameter corresponding to the number of neighbors (exploring range) was set to the interval (*Mumtaz et al., 2017*; *Bø, 2004*). LR and SVM both have a hyperparameter that corresponds to the inverse of the regularization strength (C parameter), the scanning ranges of which were set to [0.0001, 0.001, 0.005, 0.01, 0.05, 0.1] and [0.0001, 0.001, 0.005, 0.01, 0.05], respectively. Then, with exception of the SVM
- N: number of samples

- $X$: sample matrix (each row being one sample)

1. For k = 1 to N, keep $X_{\neq k}$ for model selection and $X_{=k}$ for keeping track of selected models' performance

2. For each of the ranking criteria, Pearson and RELIEF, determine the subsets of dimensions 2 through 10, using the branch and bound algorithm

3. For each subset dimension, determine the best model configuration for each classifier type (inner LOO)

4. Rank the pre-selected models on the previous step, then fit the best one over the $X_{\neq k}$ samples

5. Classify the $X_{=k}$ (outer LOO)

**Figure 3   Steps for evaluation and selection of the best configuration for each classifier.**

configuration, which had its kernel type fixed to linear, no other default settings of the scikit-learn models were overridden.

Different SVM kernels were initially tested, without a conclusive result of which would be better. Thus, the kernel type of SVM was fixed to linear to reduce the computational cost in searching the better kernel during the hyperparameters selection, besides being less prone to overfitting the data.

Both the evaluation and selection of the configuration were based on leave-one-out (LOO) cross-validation (*Längkvist, Karlsson & Loutfi, 2014*; *Shao, Meng & Wang, 2016*). This procedure consisted of two nested LOO stages (Fig. 3), with the inner one used for ordering the models in terms of validation accuracy, whereas the outer one tracked the performance of the best of all three classifiers as well as their individual best configurations.

The whole procedure was repeated for each of the four data acquisition maneuvers, with and without the subset search (Fig. 3, Step 2). Moreover, a combined search scheme using all data acquisition maneuvers was performed; prior to Step 2, each activity underwent the ranking and branch and bound methods, with the output subset dimension fixed at 6. Then, the subsets of each maneuver were concatenated, thus constituting a 24-features subset, which was then fed to the remaining selection procedure.

Accuracy ((True positive + True negative)/all subjects), precision (True positive/(True positive + False positive)), and recall (True positive/(True positive + False negative)) for each model are presented. In this study, accuracy and recall should be carefully checked because there is a high cost associated with patients set as False Negative.

The non-parametric McNemar's test (*Stapor, 2017*) was used to check for statistically significant differences ($p \leq 0.05$) between the test accuracies of the three classifiers (*Häfner et al., 2009*) with FSS application. Additionally, the Shapiro–Wilk (for normal distribution testing) and Mann–Whitney $U$ tests were applied to the distributions of the features that presented the highest selection frequencies, as selected by the combined search scheme.
Finally, a PCA was applied on the combined selection scheme with the aim of visualizing the separability between the two classes' (continent and incontinent) for two situations, using all features of the four maneuvers and using only the features that presented the highest selection frequencies.

## RESULTS

### Individual maneuvers search

For the endurance maneuver features the highest accuracy was 68.8%, which was obtained by both the best configurations out of the three classifiers and the best configuration of the LR. The highest precision was 70.6%, which was obtained by the best configuration of the k-NN without FSS, and the highest recall was 83.3%, which was obtained by the best configuration out of the three classifiers with FSS (Table 1). LR reached the highest selection frequency for this maneuver, being selected 42 times out of 48 according to the validation accuracy estimated by the inner LOO (Fig. 3), regardless of the FSS application. In addition, with the exception of the k-NN classifier, the FSS application increased the accuracy, the precision, and the recall achieved by the best configuration of each classifier. On the other hand, with the exception of the SVM classifier, the RFE application decreased the accuracy, the precision, and the recall achieved by the best configuration of each classifier (Table 1).

For the Valsalva maneuver features, the highest accuracy achieved was 58.3%, which was obtained by both the best configuration out of the three classifiers without FSS and the best configuration of the LR. The highest precision was 77.8%, which was obtained by the best configuration of the LR with RFE, and the highest recall was 58.3%, which was obtained by the best configuration out of the three classifiers without FSS, LR without FSS, and LR with RFE (Table 1). LR also reached the highest selection frequency for this maneuver, being selected 43 times out of 48 without FSS, although k-NN was selected 40 times out of 48 with RFE. In addition, the application of FSS contributed to an overall accuracy, precision, and recall decrease, except for the k-NN classifier, which had a slight increase. On the other hand, with the exception of the best configuration out of the three classifiers, the RFE application produced an overall accuracy, precision, and recall increase (Table 1).

The highest accuracy achieved for the maximum contraction maneuver features was 60.4%, which was only reached without FSS. This value corresponds to the performance obtained by the best configurations of SVM and LR, with the latter being selected by the inner LOO 48 times out of 48. The highest precision was 60.0%, which was obtained by the best configuration of the SVM without FSS, and the highest recall was 66.7%, which was obtained by the best configuration out of the three classifiers without FSS, LR without FSS, and LR with RFE (Table 1). When the FSS and RFE procedures were employed, however, k-NN reached the highest selection frequency for this maneuver, being selected 40 times out of 48 and 38 times out of 48, respectively. However, both FSS and RFE also contributed to an accuracy, precision, and recall decrease for all classifiers (Table 1).

The highest accuracy achieved for the wave maneuver features was 79.2%, which was obtained only by the best configurations of the k-NN classifier with RFE application. This classifier, presented the highest selection frequency, being selected 43 times out of

**Table 1** Selection frequency and test accuracy, precision and recall of best model out of the three classifiers and best model of each classifier, obtained with and without FSS application over the individual features of each activity.

| | | | Selected | Selected k-NN | Selected LR | Selected SVM |
|---|---|---|---|---|---|---|
| **Endurance task** | **Without FSS** | Accuracy | 0.563 | 0.646 | 0.667 | 0.542 |
| | | Precision | 0.579 | 0.706 | 0.667 | 0.533 |
| | | Recall | 0.458 | 0.500 | 0.667 | 0.667 |
| | | Sel. Frequency | – | 6/48 | 42/48 | 0/48 |
| | **With BB** | Accuracy | 0.688 | 0.458 | 0.688 | 0.646 |
| | | Precision | 0.645 | 0.458 | 0.655 | 0.629 |
| | | Recall | 0.833 | 0.458 | 0.792 | 0.708 |
| | | Sel. Frequency | – | 5/48 | 42/48 | 1/48 |
| | **With RFE** | Accuracy | 0.542 | 0.563 | 0.583 | 0.583 |
| | | Precision | 0.550 | 0.579 | 0.571 | 0.571 |
| | | Recall | 0.458 | 0.458 | 0.667 | 0.667 |
| | | Sel. Frequency | – | 36/48 | 6/48 | 6/48 |
| **Valsava Maneuver** | **Without FSS** | Accuracy | 0.583 | 0.417 | 0.583 | 0.542 |
| | | Precision | 0.583 | 0.389 | 0.583 | 0.545 |
| | | Recall | 0.583 | 0.292 | 0.583 | 0.500 |
| | | Sel. Frequency | – | 0/48 | 43/48 | 5/48 |
| | **With BB** | Accuracy | 0.500 | 0.438 | 0.583 | 0.521 |
| | | Precision | 0.500 | 0.421 | 0.611 | 0.533 |
| | | Recall | 0.333 | 0.333 | 0.458 | 0.333 |
| | | Sel. Frequency | – | 8/48 | 28/48 | 12/48 |
| | **With RFE** | Accuracy | 0.563 | 0.563 | 0.708 | 0.604 |
| | | Precision | 0.579 | 0.579 | 0.778 | 0.647 |
| | | Recall | 0.458 | 0.458 | 0.583 | 0.458 |
| | | Sel. Frequency | – | 40/48 | 5/48 | 3/48 |
| **Maximum Contraction** | **Without FSS** | Accuracy | 0.604 | 0.542 | 0.604 | 0.604 |
| | | Precision | 0.593 | 0.542 | 0.593 | 0.600 |
| | | Recall | 0.667 | 0.542 | 0.667 | 0.625 |
| | | Sel. Frequency | – | 0/48 | 48/48 | 0/48 |
| | **With BB** | Accuracy | 0.500 | 0.500 | 0.562 | 0.562 |
| | | Precision | 0.500 | 0.500 | 0.560 | 0.556 |
| | | Recall | 0.500 | 0.500 | 0.583 | 0.625 |
| | | Sel. Frequency | – | 40/48 | 6/48 | 2/48 |
| | **With RFE** | Accuracy | 0.563 | 0.521 | 0.604 | 0.563 |
| | | Precision | 0.565 | 0.522 | 0.593 | 0.556 |
| | | Recall | 0.542 | 0.500 | 0.667 | 0.625 |
| | | Sel. Frequency | – | 38/48 | 10/48 | 0 |

**Table 1** (*continued*)

|  |  |  | Selected | Selected k-NN | Selected LR | Selected SVM |
|---|---|---|---|---|---|---|
| **Wave Task** | ***Without FSS*** | *Accuracy* | 0.625 | 0.521 | 0.667 | 0.562 |
|  |  | *Precision* | 0.625 | 0.533 | 0.667 | 0.565 |
|  |  | *Recall* | 0.625 | 0.333 | 0.667 | 0.542 |
|  |  | *Sel. Frequency* | – | 0/48 | 41/48 | 7/48 |
|  | ***With BB*** | *Accuracy* | 0.708 | 0.667 | 0.688 | 0.729 |
|  |  | *Precision* | 0.708 | 0.700 | 0.680 | 0.762 |
|  |  | *Recall* | 0.708 | 0.583 | 0.708 | 0.667 |
|  |  | *Sel. Frequency* | – | 6/48 | 37/48 | 5/48 |
|  | ***With RFE*** | *Accuracy* | 0.771 | 0.792 | 0.729 | 0.667 |
|  |  | *Precision* | 0.842 | 0.889 | 0.704 | 0.654 |
|  |  | *Recall* | 0.667 | 0.667 | 0.792 | 0.704 |
|  |  | *Sel. Frequency* | – | 43/48 | 4/48 | 1/48 |

**Notes.**

k-NN, k-nearest neighbors; LR, logistic regression; SVM, support vector machine; FSS, feature subset search; BB, branch and bound; RFE, recursive feature elimination.

48 with RFE, although LR was selected 41 times out of 48 without FSS and 37 times out of 48 with FSS (Table 1). The highest precision was 88.9%, which was obtained by the best configuration of the k-NN with RFE, and the highest recall was 79.2%, which was obtained by the best configuration of the LR with RFE (Table 1). In addition, the FSS and RFE application increased the accuracy, precision, and recall achieved by the best configurations of all classifiers (Table 1).

Regarding classifier performance with BB application, for the endurance maneuver, there was significant difference between LR and k-NN performance as well as between SVM and k-NN performance (Table 2). No significant differences were observed for the other maneuvers. The McNemar test results suggest statistical inferiority of the k-NN when applied to the endurance maneuver features. Regarding classifier performance with RFE application, for the Valsava maneuver, there was significant difference between SVM and LR performance as well as between k-NN and LR performance (Table 2). No significant differences were observed for the other maneuvers. The McNemar test results suggest statistical inferiority of the LR when applied to the Valsava maneuver features.

## Combined maneuvers search

With the combined maneuvers search scheme, the highest achieved accuracy was 77.1%, which was obtained by the best configurations out of the three classifiers with FSS, the best configurations of the LR with FSS, the best configuration of the LR with RFE, and the best configuration of SVM with RFE (Table 3). The highest precision was 81.8%, which was obtained by the best configuration of the k-NN without FSS, and the highest recall was 83.3%, which was obtained by the best configuration of the LR with RFE, and the best configuration of SVM with RFE (Table 3). K-NN attained the highest selection frequency amongst the classifiers, being selected 41 times out of 48 with RFE. Overall, the FSS and RFE application increased the accuracy, precision, and recall achieved by the best

**Table 2  McNemar's *p*-values for the comparison of performance obtained by the three classifiers, on each activity, as well as combined strategies, without FSS, with BB and with RFE application.**

|  |  | SVM vs k-NN | SVM vs LR | k-NN vs LR |
|---|---|---|---|---|
| Without FSS | Endurance Task | 0.197 | 0.014* | 0.782 |
|  | Valsava Maneuver | 0.083 | 0.479 | 0.046 |
|  | Maximum Contraction | 0.467 | 1.0 | 0.467 |
|  | Wave Task | 0.617 | 0.025* | 0.089 |
|  | Combined | 0.637 | 0.414 | 1.0 |
| With BB | Endurance Task | 0.029* | 0.157 | 0.007* |
|  | Valsava Maneuver | 0.248 | 0.257 | 0.089 |
|  | Maximum Contraction | 0.317 | 1.0 | 0.083 |
|  | Wave Task | 0.365 | 0.414 | 0.739 |
|  | Combined | 0.285 | 0.096 | 0.077 |
| With RFE | Endurance Task | 0.763 | 1.0 | 0.781 |
|  | Valsava Maneuver | 0.414 | 0.025* | 0.019* |
|  | Maximum Contraction | 0.317 | 1.0 | 0.083 |
|  | Wave Task | 0.527 | 0.157 | 0.206 |
|  | Combined | 0.655 | 1.0 | 0.739 |

**Notes.**
\* indicates significant differences.
k-NN, k-nearest neighbors; LR, logistic regression; SVM, support vector machine; FSS, feature subset search; BB, branch and bound; RFE, recursive feature elimination.

configurations of all classifiers, with the greatest increase for the LR (Table 3). Moreover, no statistical differences were observed between the classifiers' performances (Table 4).

From the six extracted features that presented the highest selection frequencies, i.e., the most frequently selected features, significant differences between the classes (continent vs. incontinent women) were observed (Mann–Whitney U test, $p < 0.05$), except for the instant of activation of the sensor grouping $Ring_2$ extracted from the Valsalva maneuver (Fig. 4). With these six features, the class separability gain, observed through the two-principal component plot, is quite visible if compared to the one attained with all features of the four activities (Figs. 5 and 6).

When applying the configuration selection procedure (Fig. 3) only on these six more frequently selected features with no further FSS, it was possible to achieve accuracy as high as 97.9% for the best configurations of the LR and SVM classifiers as well as 95.8% for the best configuration out of the three classifiers.

**Table 3** Selection frequency and test accuracy, precision, and recall of best model out of the three classifiers and best model of each classifier, obtained without FSS, with BB, and with RFE application over the combined activities search scheme.

|  |  | Selected | Selected k-NN | Selected Logistic R. | Selected SVM |
|---|---|---|---|---|---|
| **Without FSS** | *Accuracy* | 0.667 | 0.646 | 0.646 | 0.688 |
| | *Precision* | 0.667 | 0.818 | 0.652 | 0.680 |
| | *Recall* | 0.667 | 0.375 | 0.625 | 0.708 |
| | *Sel. frequency* | – | 1/48 | 27/48 | 20/48 |
| **With BB** | *Accuracy* | 0.771 | 0.750 | 0.771 | 0.667 |
| | *Precision* | 0.809 | 0.773 | 0.783 | 0.682 |
| | *Recall* | 0.708 | 0.708 | 0.750 | 0.625 |
| | *Sel. frequency* | – | 0/48 | 31/48 | 17/48 |
| **With RFE** | *Accuracy* | 0.750 | 0.750 | 0.771 | 0.771 |
| | *Precision* | 0.773 | 0.773 | 0.741 | 0.741 |
| | *Recall* | 0.708 | 0.708 | 0.833 | 0.833 |
| | *Sel. frequency* | – | 41/48 | 6/48 | 1/48 |

Notes.
k-NN, k-nearest neighbors; LR, logistic regression; SVM, support vector machine; FSS, feature subset search; BB, branch and bound; RFE, recursive feature elimination.

**Table 4** McNemar's *p*-values for the comparison of performance obtained by the three classifiers on the combined search scheme.

| Comparison | *SVM vs k-NN* | *SVM vs LR* | *k-NN vs LR* |
|---|---|---|---|
| *p*-value | 0.285 | 0.096 | 0.782 |

Notes.
k-NN, k-nearest neighbors; LR, logistic regression; SVM, support vector machine.

Finally, the test accuracies of the best models out of the three classifiers are presented in Table 5.

## DISCUSSION

The present study aimed to evaluate the potential of an automatic diagnostic system for discrimination between women with and without UI through data collected by a novel intravaginal pressure probe (*Cacciari et al., 2017a*). Although the k-NN with RFE and wave task data produced the best accuracy (79.2%) and precision (88.9%), it reached a poor recall (66.7%) (Table 1) that is imperative in this study. Thus, the results indicated that, overall, the best performance was achieved by combining the features of all four acquired maneuvers (SVM with RFE: 77.1% accuracy, 74.1% precision, and 83.3% recall—Table 3) meaning that UI is a combination of failures in pelvic floor functioning. Moreover, despite not reaching levels of performance required by an automatic diagnosis, the results provided insights into the major aspects of PFM functioning that may help in discriminating continent from incontinent women.

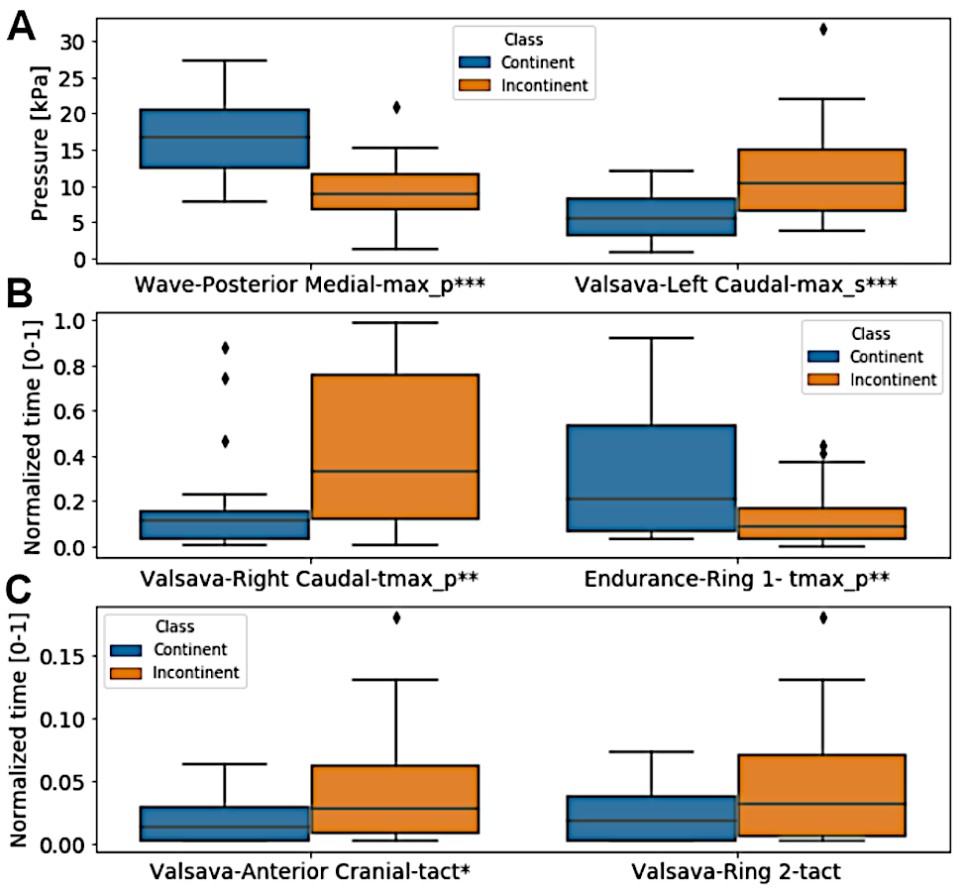

**Figure 4 Box plot of the six extracted features that presented the highest selection frequencies comparing the two classes, continent vs incontinent women.** (A) pressure for wave-posterior medial-max_p and valsava-left caudal-max_s; (B) normalized time for valsava-right caudal-tmax_p and endurance-ring 1-tmax_p; (C) normalized time for valsava-anterior cranial-tact and valsava-ring 2-tact. ***$p < 0.001$; **$p = 0.01$; *$p = 0.028$. Only Valsalva Ring$_2$ did not present a significant difference (Mann–Whitney $U$ test, $p > 0.05$).

In a previous study, the k-means, discriminant analysis, LR, decision tree, and two genetic algorithms were compared when diagnosing three types of incontinence (*Laurikkala et al., 1999*), including stress UI. The data were categorical and obtained through clinical questionnaires and exams. The number of samples per class was 323 incontinent and 207 continent women. Despite the differences in sample size and data properties compared to the present study, the only statistically significant difference observed also pointed to the inferiority of the neighborhood voting classifiers (see Table 2). The advantage of analyzing the spatiotemporal pressure profile of the pelvic floor is that it can reveal the mechanistic characterization of the UI compared to only clinical features extracted from questionnaires.

Despite the absence of consistent statistical differences among the three classifiers, the selection frequency was higher for LR without FSS and with BB, with the exception of the maximum contraction maneuver, in which k-NN held the majority of selections when using BB. The reason for this LR predominance is not completely evident, but it is likely due

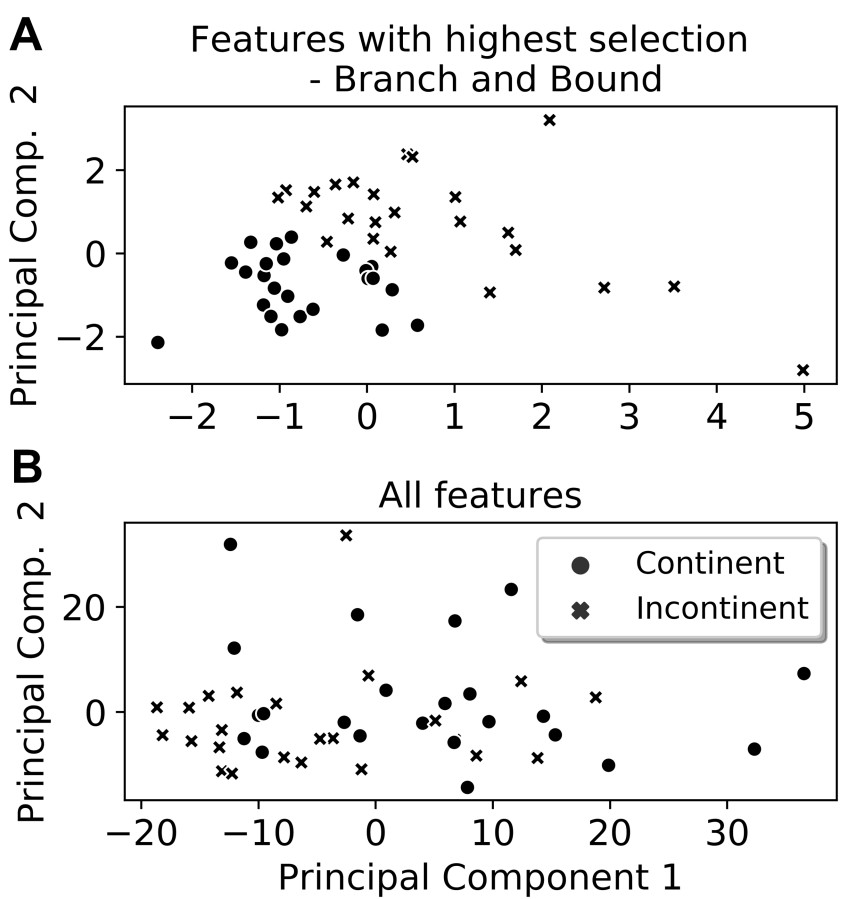

**Figure 5** **Two-principal components plot applied to the features with the highest selection frequency (A) using Branch and Bound and with all extracted features (B).** Observe that the separation between the classes is more visible when using the features with the highest selection frequencies (A).

to the low complexity of its decision rules composition process or even due to a beneficial bias introduced by either the ranking criteria or the BB process. However, with RFE, k-NN reached the highest selection frequency, although lower than LR without FSS and with BB.

Overall, the application of FSS increased the accuracy of the best-selected configurations for all maneuvers. The cases in which its application occasioned a decrease in performance, this corresponded to the two maneuvers with the lowest test accuracies (Table 5). The results obtained with the Valsalva maneuver and maximum contraction maneuver were not able to surpass the accuracy achievable by a fair coin toss, for example. Despite the low performance, Valsalva maneuver features played a major role (see Fig. 4) in the accuracy increase observed with the FSS application on the combined features search scheme (Table 5, last column). Regarding the non-discriminating capacity of the endurance maneuver in identifying incontinent women, which is a common test performed in clinical practice, , our results show that the endurance maneuver might not be the best test to identify failures in the pelvic floor of women with UI. This is in agreement with the most recent guidelines for UI (*Dumoulin, Cacciari & Hay-Smith, 2018*; *Bø& Sherburn, 2005*), in

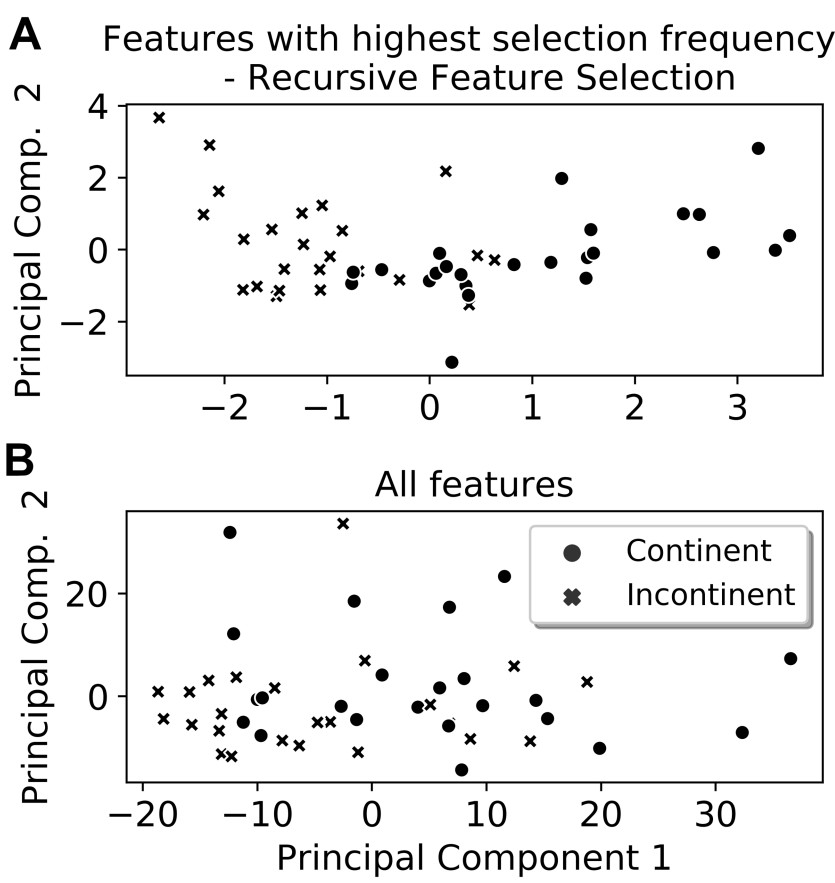

**Figure 6** Two-principal components plot applied to the features with the highest selection frequency (A) using Recursive Feature Elimination and with all extracted features (B). Observe that the separation between the classes is more visible when using the features with the highest selection frequencies (A).

**Table 5 Test accuracies of the best models out of the three classifiers, without FSS, with BB, and with RFE application.**

|  | Endurance Task | Valsava Maneuver | Maximum Contraction | Wave Task | Combined Search |
|---|---|---|---|---|---|
| **Without FSS** | 0.563 | 0.583 | 0.604 | 0.625 | 0.667 |
| **With BB** | 0.688 | 0.500 | 0.500 | 0.708 | 0.771 |
| **With RFE** | 0.542 | 0.563 | 0.563 | 0.771 | 0.750 |

**Notes.**
FSS, feature subset search; BB, branch and bound; RFE, recursive feature elimination.

which coordination and asymmetry may play a major role in pelvic floor dysfunctions, and support the good results with wave maneuver (except for recall value).

Three of the Valsalva maneuver features are among the group of six variables with the highest selection frequency (Fig. 4). Besides increasing inter-class separability (Fig. 5), this also represents two important aspects of the analysis of PFM functioning, namely strength generation (peak values of pressure) and coordination (time instants) (*Bø& Sherburn,*

*2005*). The instant of maximum pressure, for example, of the incontinent group during the Valsalva maneuver was significantly higher than the one observed in the continent group (Fig. 4), which may suggest lack of coordination in the counter-response to the intra-abdominal pressure increase, hence leading to an involuntary urine leakage (*Wyndaele & Abrams, 2018*).

This study has some limitations. The major concern is the relatively small data sample used for automatic classification of UI, which would not be sufficient to provide reliable results. However, it was not our intention to provide a final and finished automatic classification system, but to show the potential use of pelvic floor pressure distribution profiles in its construction as an objective assessment of UI, and to test some feature selection algorithm and some classifiers. In addition, the equipment for pelvic floor pressure distribution data collection is novel, relatively expensive, and can only be found in research laboratories. Thus, with a larger data sample, other feature selection algorithms, other classifiers, and even automatic feature extraction should be tested.

On the other hand, the novel equipment, although not essential for UI assessment in a clinical context, not substituting or overcoming other clinical invasive measurements, provides information for distinguishing vaginal sub-regions, planes, rings and maneuvers, contributing to evaluate aspects such as coordination and asymmetry that may play an important role in pelvic floor dysfunctions.

Finally, the performance attained using the six features with the highest selection frequencies, which were able to reach 97.9% accuracy, require further validation with a new and larger data set. This results in the 77.1% accuracy as the highest test accuracy achieved in the present study, demonstrating the potential for an automatic diagnosis system for discriminating female UI using quantitative intravaginal pressure data.

This result is quite below the 96% accuracy obtained by the discriminant analysis model in the study where two other types of UI were also classified (*Laurikkala et al., 1999*). In addition to the fact that the data used by Laurikkala et al. were categorical, it is not clear if the reported performance was a test or validation accuracy, which, in the latter case, would provide an over-optimistic performance metric.

## CONCLUSIONS

This first attempt to use intravaginal pressure data to automatically diagnose female UI demonstrated that the data have discriminatory potential depending on how well they are harnessed. When fed with all features, the employed RFE algorithm was able to produce the best-achieved accuracy and, although not at the level required by an automatic system, provided insights into the PFM functioning aspects contributing to a UI diagnosis. Overall, the best performance was achieved by combining the features of all four acquired maneuvers. When considering the group of six variables with the highest selection frequency, the Valsalva maneuver had the greatest impact. Further, the endurance maneuver might not be an advisable test for UI classification, and wave maneuver produced good results, except for recall value.

To further explore the data potential, a larger data set is necessary since the use of an intravaginal probe for pressure data collection is a very new tool for incontinence urinary

assessment in women. Additionally, a reduction in the branch and bound computational cost is needed in order to increase the number of features kept after the first ranking stage. Furthermore, the application of other methods of feature extraction, including supervised methods based on neural networks, should further test the potential of intravaginal pressure data for classification.

### Funding
This project was funded by Sao Paulo Research Foundation (FAPESP 2013/19610-3). Adriano Carafini's scholarship was provided by the Foundation for Research Support of State of Goiás (FAPEG), Brazil (Process: 201710267000687). Isabel Sacco and Marcus Vieira are fellows of the National Council for Scientific and Technological Development (CNPq), Brazil (Process: 304124/2018-4 and 306205/2017-3, respectively). There was no additional external funding received for this study. The funders had no role in study design, data collection and analysis, decision to publish, or preparation of the manuscript.

### Grant Disclosures
The following grant information was disclosed by the authors:
Sao Paulo Research Foundation: FAPESP 2013/19610-3.
Research Support of State of Goiás (FAPEG), Brazil: 201710267000687.
National Council for Scientific and Technological Development (CNPq), Brazil: 304124/2018-4, 306205/2017-3.

### Competing Interests
The authors declare there are no competing interests.

### Author Contributions
- Adriano Carafini conceived and designed the experiments, performed the experiments, analyzed the data, prepared figures and/or tables, authored or reviewed drafts of the paper, approved the final draft.
- Isabel C.N. Sacco conceived and designed the experiments, authored or reviewed drafts of the paper, approved the final draft, provided data and information about equipment, experimental protocols and data collection.
- Marcus Fraga Vieira conceived and designed the experiments, analyzed the data, prepared figures and/or tables, authored or reviewed drafts of the paper, approved the final draft.

### Human Ethics
The following information was supplied relating to ethical approvals (i.e., approving body and any reference numbers):
The Ethics Committee of the School of Medicine of the University of São Paulo approved this research (protocol n.023/14).

## Data Availability

Data and codes are available at Zenodo: Carafini, Adriano. (2019). Female Urinary Incontinence Classification Code [Data set]. Zenodo. http://doi.org/10.5281/zenodo.3268009.

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
