# Peer review of "Pelvic floor pressure distribution profile in urinary incontinence: a classification study with feature selection"

_PeerJ, doi:10.7717/peerj.8207_

## Round 0.1 · original submission · Major Revisions

Your manuscript has been reviewed by two experts. The first reviewer is an expert in machine learning while the second reviewer is a medical doctor specialized in the field. As you can see from their comments below, both of them raise substantial points. Particularly, since the data size used in this study is not so large, the evaluation of the reliability of their results is quite essential. More discussion on the obtained results from medical perspective would be one way in this respect. Please read their comments carefully and revise the manuscript accordingly.

Reviewer 1 ·

Basic reporting

In this paper, they establish the group of the features that provides the greatest capability between continent and incontinent women. To achieve this, they combined a ranking method and a feature subset method. Overall, I didn’t see much novelty of this paper. Moreover, the other critical problem is that the dataset is too small to build a reliable model. Using such small dataset for performance evaluation, I doubt the results they presented are not reliable either. In section of results, they presented the feature comparison between before and after feature selection. This is not enough. What they should do more is to compare other feature selection algorithms.

Experimental design

More comparison is needed.

Validity of the findings

No much novelty is found in the paper.

Additional comments

In this paper, they establish the group of the features that provides the greatest capability between continent and incontinent women. To achieve this, they combined a ranking method and a feature subset method. Overall, I didn’t see much novelty of this paper. Moreover, the other critical problem is that the dataset is too small to build a reliable model. Using such small dataset for performance evaluation, I doubt the results they presented are not reliable either. In section of results, they presented the feature comparison between before and after feature selection. This is not enough. What they should do more is to compare other feature selection algorithms. I have some concerns as follows.

Major comments:

In abstract, it is not clear to see the contribution of this paper. I suggest them to rewrite this section and highlight the contribution of this paper.

In Feature extraction, rather than describing how to compute the features, the authors should clarify why they choose those features. Is there any evidence to explain the importance of the features for classification in this study?

The result comparison between before feature selection and after the feature selection is not enough. To explain the superiority of their feature selection method, they should compare their feature selection method
with different feature selection methods, like using a combination of different ranking (like and feature subset searching methods.

In dataset, I doubt the reliability of the results they presented, because the datasets are too small to build a reliable model and evaluate the classification performance.

For performance evaluation, the accuracy is an important metric for evaluating the overall performance of a model, but they still need to include more other evaluation metrics, like sensitivity, specificity, or auROC, which are all important to comprehensively evaluate the model’s performance .

Minor comments:

It’s not clear why they choose the Mahalanobis distance as the subset evaluation metric, rather than others, like typical Euler distance.

Why fixing the kernel type of SVM to linear.

·

Basic reporting

This report is the first attempt to establish the group of features that provides the greatest discrimination capability between continent and incontinent women in order to use intravaginal pressure data to automatically diagnose female UI.
Although the methods of extracting data from four clinically used maneuvers and analyzing three classifiers with and without FSS are interesting and unique, it is necessary to discuss whether this system can be put to practical use in urology outpatients. In general, by asking the patient's complaints, it is almost clear whether the patient has urinary incontinence or continence.
The following points should be considered and revised.

Experimental design

1. The diagnosis of urinary incontinence in women begins with questions and questionnaires such as when and how much urine leaks. In fact, the diagnoses of the urinary continence group, UI(stress UI, urge UI, and mixed UI) are almost possible at this point. Moreover, the urodynamics study that is often used clinically as an objective test can also assess the degree of intravesical pressure and involuntary contraction, and diagnose the severity of UI. The authors should describe the potential benefits for outpatient clinics in the future, if applying this methodology and device used in this study.
2. The factor that determines the degree of urinary incontinence are considered simply the balance of intravesical pressure to urinate and urethral sphincter pressure to retain it. Although urinary sphincter pressure and intravaginal pressure may be correlated, the data measured on the patient side are only vaginal pressure by four maneuvers. Please provide some rationales that the authors did not need to consider the other measurable elements, directly intravesical pressure and abdominal pressure (rectal pressure).

Validity of the findings

3. Although the six extracted features effective were selected in Fig. 4, it should be explained and considered from the clinical significance why they are significantly different between continent and incontinent.
4. It is difficult to understand how 97.9% and 95.8% are calculated in the lines 313-314 of Results. Please explain this way of calculation.
5. Please mention some of the limitations in this study in a separate paragraph of Discussion.

---

## Round 0.2 · accepted · Accept

Your revised manuscript has been reviewed by the same two original reviewers. As you can find from their comments below, both of them admit that the manuscript has been appropriately revised (though one of them, a medical doctor in the field, is still skeptic with its medical significance). Thus, I am happy to inform you that I recommend its acceptance.

Reviewer 1 ·

Basic reporting

No comment

Experimental design

No comment

Validity of the findings

No comment

Additional comments

All my concerns have been roughly addressed.

·

Basic reporting

From the medical point of view, I think that this study is not immediately usable in clinical practice, but it has been appropriately commented and revised, including the points pointed out by other reviewers.

Experimental design

Because of the fundamentals of this study, although not necessarily a complete revision, the two points I pointed out have been appropriately commented and revised.

Validity of the findings

The three points I pointed out are appropriately commented and revised.

Additional comments

From the medical point of view, I think that this study is not immediately usable in clinical practice, but it has been appropriately commented and revised, including the points pointed out by other reviewers.